## [Transparent Peer Review file · Nature Communications]

Hepatocytes Functionally Reprogrammed by KIAA1199-High Colorectal Cancer Cells Favour the Accumulation of Pro-metastatic Egr1+ Neutrophils

Corresponding Author: Dr dejun zhang

Version 0:

Reviewer comments:

Reviewer #1

(Remarks to the Author)

This study by Lisha Li et al. investigates the role of KIAA1199 in promoting colorectal cancer derived liver metastasis (CRLM). The authors propose that tumor-derived KIAA1199 suppresses hepatocyte PPAR γ , leading to SAA2 induction, which activates neutrophil FPR2 signaling and promotes the emergence of EGR1⁺ pro-angiogenic tumor-associated neutrophils (TANs) at the tumor–liver interface. The study combines orthotopic and intrasplenic CRC mouse models with spatial transcriptomics, flow cytometry, functional assays, and serum biomarker analyses. The novelty of the work is the mechanistic axis proposed. Hepatocyte SAA priming of the liver niche has been shown previously and SAA promoting FPR2 signaling is established. However, the identification of PPAR γ as an upstream brake on hepatocyte SAA2 expression, the characterization of an EGR1⁺ neutrophil state that promotes angiogenesis at the tumor–liver interface and the proposed translational biomarker model (KIAA1199+SAA2 in serum) are novel. These contributions are valuable. However, there are several mechanistic and methodological weaknesses that make it hard to fully evaluate this work.

Major points

1. Flow cytometry: Absolute numbers in addition to percentages should be provided. Furthermore, supp. figures detailing the gating strategy should be added.
2. Imaging inconsistency: It is not clear, if MRI or CT was used, e.g. for Fig2 there is an inconsistency between the Figure and the description in the results. In any case, the acquisition parameters should be clearly stated (scanner used, was contrast reagent administered or not, analysis workflow).
3. Statistical tests: The used test should be clearly stated in each figure legend and the number of replicates should be indicated. Please also indicate how often the experiment was repeated. I would consider using non-parametric tests, and also tests accounting for multiple testing for some experiments.
4. Figure 1M: shControl appears to be associated with lower survival; please verify labels/analysis. If this is indeed the case, then possible reasons should be discussed.
5. Regarding the single-cell analysis, the authors refer to liver-infiltrating cells. Details should be provided to understand, if these are indeed liver-infiltrating cells.
6. EGR1+ neutrophil fractions appear to vary between figures. Is this due to different genetic background of the mice used? In any case, the used genetic background should be clearly stated in each figure.
7. ChIP-qPCR appears to be run in CRC lines and not hepatocytes. Since this leads to one major conclusion of the study, I feel that hepatocyte-specific evidence is essential.
8. There is an inconsistency regarding the Rosiglitazone dose. In Figure 4E 5 mg/kg/day i.p. was used, while in Figure 4N 10 μ g/kg/day i.p. was used for apparently the same intervention. This should be clarified.
10. Several experiments (e.g., Fig. 5C) report very low, and in some cases zero, metastatic lesions. This raises technical concerns how reliable the intrasplenic model was in their hands. It would be important to clarify whether this reflects biological heterogeneity or technical variation. In this regard, the description of the intrasplenic injection protocol appears to differ from published standards such as STAR Protocols (e.g. STAR Protocol 3250). Variations in technique can markedly influence metastatic take rate, extrahepatic spillover, and reproducibility. Please provide references for the protocol used.

Also, please include methodological details and overall success rates to allow comparison with established methods.

Minor points

1. provide presentative images for the scratch wound healing assay

Reviewer #2

(Remarks to the Author)

This study provides novel mechanistic insights into pre-metastatic niche (PMN) formation in colorectal cancer liver metastasis (CRLM). The authors identify and rigorously validate a previously unrecognized "PPAR γ /SAA2/FPR2-EGR1" signaling axis, driven by tumor cell KIAA1199 overexpression, which orchestrates critical tumor cell-hepatocyte-neutrophil cross-talk to establish the metastatic-prone liver microenvironment. The authors employ an orthogonal dual-tumor model (cecum/spleen) to mimic primary tumor-driven PMN priming. Using comprehensive multi-omics analyses (scRNA-seq, spatial transcriptomics, pseudotime, pySCENIC, CellChat), it maps a key axis, which is validated through extensive functional experiments (gain/loss-of-function targeting KIAA1199/EGR1, agonists/antagonists like Rosiglitazone/WRW4/Wortmannin, exogenous SAA2). Strong clinical relevance is also demonstrated via significant correlation of core axis components (KIAA1199, SAA2) with patient outcomes across TCGA, an internal cohort (tissue/serum), and an independent validation cohort.

Major Concerns:

1. The article presents a fundamental unanswered question regarding how KIAA1199 transmits signals from tumor cells to hepatocytes, which is the starting point of the entire signaling axis. This mechanism is the most vaguely elucidated. While the article mentions that KIAA1199 can be secreted and detected in patient serum, correlating with tissue expression, it remains unclear whether secreted KIAA1199 directly acts on distal hepatocytes or if its effects are indirect, mediated by other consequences of KIAA1199 overexpression (e.g., induction of other factors). Are there other carriers involved (e.g., exosomes)? Does it act indirectly by influencing systemic inflammatory factors? It is recommended to supplement experiments, such as treating hepatocytes with conditioned medium (before and after exosome depletion) or purified recombinant KIAA1199 protein, to assess whether the phenotypes can be recapitulated. This critical mechanistic ambiguity should be thoroughly analyzed and discussed within the manuscript.
2. Is there a specific receptor or binding partner for KIAA1199 or its related products (e.g. LMW-HA) on hepatocytes? Relatedly, to functionally test whether any secreted KIAA1199 reaching hepatocytes acts directly through such an interaction, it would be valuable to determine if a neutralizing or blocking antibody against KIAA1199 or its binding partner can abrogate its effects. If such an antibody is available (or could be developed/generated), experiments treating hepatocytes with KIAA1199-containing conditioned medium or recombinant protein in the presence of this blocking antibody would provide crucial evidence. A lack of phenotype rescue by the antibody would support an indirect mechanism, while effective blocking would strongly implicate a direct interaction requiring a specific KIAA1199 receptor/binding partner on hepatocytes. Investigating potential hepatocyte surface receptors should be a priority.
3. The naming "Malignant-prone" is eye-catching but potentially inadequately supported by the data. The data show these hepatocytes have increased CNV and metabolic reprogramming, but there is no direct evidence that they themselves transform into malignant cancer cells. Their "malignant" nature is more likely indicative of their function in promoting malignant processes within the microenvironment. It is suggested to adopt more conservative terminology, such as "pre-metastatic hepatocytes" or "functionally reprogrammed hepatocytes," and include relevant analysis in the discussion.
4. Quantification of Spatial Co-localization. Figure 3E-G and Supplementary Figure 10A demonstrate the co-localization of EGR1+ neutrophils and SAA2+ hepatocytes but lack quantitative statistical analysis to support the observations.
5. Multiplex Staining Validation. For Figures 2 and 3, it is recommended to supplement with multiplex staining images (e.g., IMC or mIF) showing tumor cell markers, hepatocyte markers, SAA2, and EGR1 within the same tissue locations to solidify the spatial relationships depicted in the scRNA-seq and spatial transcriptomic analyses.

Minor Concerns:

1. The use of the PPAR γ agonist Rosiglitazone (Figure 4N-S) carries known clinical cardiovascular safety concerns that led to prescribing restrictions. The study does not address or discuss the dose used relative to known safety thresholds or the translational implications of these risks.
2. Clarify the specific statistical method used for each type of data analysis throughout the manuscript. Ensure that figure legends explicitly state the sample size (n) for each experiment/group shown in the figure.
3. In Figure 2 and 3, specifically in the immunofluorescence staining panels, the font colors labeling specific protein markers (e.g., Ly6g) do not match the actual staining colors shown in the images. For instance, Ly6g is labeled in light green font, but the actual staining appears deep green. Ensure label colors accurately represent the fluorophores used.

Reviewer #3

(Remarks to the Author)

In this manuscript, Lisha Li et al. investigate how KIAA1199-high colorectal cancer (CRC) tumors orchestrate the hepatic pre-metastatic niche (PMN) through malignant-prone hepatocytes and EGR1⁺ neutrophils, ultimately promoting liver metastasis. The study is ambitious, methodologically diverse, and overall very well written and designed. The authors provide compelling evidence for a PPAR γ /SAA2/FPR2-EGR1 axis that links hepatocyte signaling to neutrophil reprogramming and angiogenesis.

Main Concerns

The authors inject CRC cells into the spleen while the primary orthotopic tumor remains in place. This raises the possibility that primary tumor cells continue to disseminate and contribute to metastatic burden, confounding the interpretation of “experimental” versus “spontaneous” metastasis. The authors could use RFP/GFP labeled cells to distinguish between which cells are in the liver.

This concern is particularly important because EGR1⁺ neutrophils promote vascular remodeling, which could in turn facilitate additional extravasation of primary tumor cells.

If technically feasible, resection of the primary CRC prior to splenic injection would strengthen the conclusions.

The authors should provide evidence that no cancer cells (including dormant or isolated cells) are already present in the liver before splenic injection. Otherwise, it is difficult to distinguish metastases seeded earlier from those generated experimentally.

Specific Points

- 1) The authors should show KIAA1199 expression in the CRC cell lines used. Also, it would have been nice if the authors used cell lines with different KIAA1199 expression levels instead of modifying the levels experimentally.
- 2) In Fig. 1G–H, the growth and size of the primary CRC tumors upon modulation of KIAA1199 should be reported.
- 3) In Fig. 1G, it is essential to demonstrate the absence of cancer cells, even dormant or isolated ones, in the liver prior to splenic injection. Without this control, it is not possible to distinguish spontaneous metastasis from experimental metastasis in subsequent analyses (Fig. 1H–M).
- 4) In Fig. 1G, it would also be informative to perform splenic injections with CT26 cells expressing different levels of KIAA1199 to assess whether expression impacts the ability of cancer cells to adapt to the PMN. (this point is really optional)
- 5) In Fig. 1K, the label “tumor area percentage” should be corrected to “liver metastatic area percentage.”
- 7) In the text, line 388/389, “SF2E” should be corrected to “SF5E.”
- 8) The expression of EGR1 in neutrophils and associated gene expression (particularly the data in Fig. S5E) should be assessed in the context of shKIAA1199 models.
- 9) In Fig. S6A–F, the authors claim to show hepatic accumulation of EGR1⁺ neutrophils in CRLM patients, but EGR1 expression is not directly indicated. This requires clarification.
- 10) In Fig. 2A and 2G–H, it is unclear whether the samples correspond to a pre-metastatic liver or already metastatic tissue. These experiments are done at day 14 following orthotopic injection, while it was a day 7 post-injection in Fig. 1.
- 11) In Fig. 2K, the apparent expression of EGR1 in metastatic cancer cells should be discussed.
- 12) In Fig. 2L, HL60-derived cells are used as “neutrophil-like” cells, but they are not true neutrophils. Experiments with liver-isolated neutrophils in the ±KIAA1199 context would be more physiologically relevant. Furthermore, the rationale for analyzing endothelial tube formation should be better justified with transcriptomic data.
- 13) In Fig. 2M, the authors refer to “new vessels” in the metastatic liver without prior evidence. They should also address whether increased angiogenesis could facilitate additional seeding from the primary tumor. Use of CRC resection models would help clarify this.
- 14) In Fig. 3C, it should be confirmed whether Hepa-C11 hepatocytes are absent in shKIAA1199 models, ideally through IHC or IF validation.
- 15) The designation “malignant-prone hepatocytes” may be misleading, as it could suggest actual transformation of hepatocytes. The authors should consider rephrasing this terminology if possible.
- 16) In Fig. 3D, since Hepa-C11 cells display an angiogenic signature, their role relative to EGR1⁺ neutrophils in promoting angiogenesis should be discussed.
- 17) In Fig. 3F, the hepatocyte identity of SAA2⁺ cells could be validated with a hepatocyte-specific marker. The authors should also examine whether FPR2 expression is unique to EGR1⁺ neutrophils or broadly expressed across neutrophil subsets.
- 18) In Fig. 3H, it should be clarified from which tissue neutrophils were isolated, and whether systemic SAA2 injection induces EGR1⁺ neutrophil accumulation in tissues other than the liver.
- 19) The effect of WRW4 on primary tumors should be addressed, and it should be clarified whether EGR1⁺ neutrophils are also present in the primary tumor, or whether their induction depends on hepatocyte signaling.
- 20) Since neutrophil survival is central to the proposed mechanism, *in vivo* validation using BrdU⁺ incorporation could be valuable (see PMID: 38177532). This experiment is optional but would be a great touch to the manuscript.
- 21) In Fig. 5O–R, the conclusion that EGR1⁺ neutrophils promote angiogenesis via VEGF-A is not fully supported, as only secretion data are shown. The claim should be toned down and presented as a suggestion rather than a definitive conclusion.
- 22) In Fig. 6, it would substantially strengthen the study to directly test whether blocking VEGF-A impairs the pro-angiogenic activity of EGR1⁺ neutrophils, for example by using a neutralizing antibody.

Overall, even without answering all my comments experimentally, I think this manuscript is very strong candidate for publication in Nature Communications.

Version 1:

Reviewer comments:

Reviewer #1

(Remarks to the Author)

The authors have addressed all my comments and concerns.

Reviewer #2

(Remarks to the Author)

The authors addressed most of my concerns; however, several important methodological details and presentation elements remain insufficiently described.

Missing details for the first comment (exosome work):

The manuscript does not clearly describe the exosome isolation and identification/labeling procedures. Please specify the isolation method. And also lack a description of the PKH26 labeling method for EVs.

Lack of information on proteomics and differential analysis:

The response mentions "proteomic profiling" and "differential gene analysis," but the manuscript does not provide sufficient information on the methods and sources.

Recommendation for Result 13B (group comparison):

In Result/Figure 13B, it would be more informative to include a comparison between KIAA1199-high and KIAA1199-low groups (ideally with the grouping definition and sample size), rather than presenting aggregated results only.

Exosome purification quality control:

For purified exosomes, quality control evidence should be provided, preferably including representative images/plots. A comparison of EV quantity and morphology between the two EV groups would be better included.

In line 364, there was a punctuation error "either high or low levels of KIAA1199. when the primary tumor was established but before any spontaneous metastasis occurred,"

Reviewer #3

(Remarks to the Author)

I would like to thank the authors for carefully addressing all of my comments. I believe the manuscript is of high quality, and I am confident that readers of Nature Communications will enjoy reading it as much as I enjoyed reviewing it.

Version 2:

Reviewer comments:

Reviewer #2

(Remarks to the Author)

The authors have addressed all my concerns.

Point-to-point response to reviewers' comments and suggestions on the manuscript entitled "KIAA1199-High Colorectal Cancer-Induced Malignant-Prone Hepatocytes Drive Liver Metastasis through PPAR γ /SAA2/FPR2-Mediated EGR1⁺ Neutrophils Reprogramming".

COMMENTS TO THE AUTHOR:

Reviewer: 1

This study by Lisha Li et al. investigates the role of KIAA1199 in promoting colorectal cancer derived liver metastasis (CRLM). The authors propose that tumor-derived KIAA1199 suppresses hepatocyte PPAR γ , leading to SAA2 induction, which activates neutrophil FPR2 signaling and promotes the emergence of EGR1⁺ pro-angiogenic tumor-associated neutrophils (TANs) at the tumor–liver interface. The study combines orthotopic and intrasplenic CRC mouse models with spatial transcriptomics, flow cytometry, functional assays, and serum biomarker analyses. The novelty of the work is the mechanistic axis proposed. Hepatocyte SAA priming of the liver niche has been shown previously and SAA promoting FPR2 signaling is established. However, the identification of PPAR γ as an upstream brake on hepatocyte SAA2 expression, the characterization of an EGR1⁺ neutrophil state that promotes angiogenesis at the tumor–liver interface and the proposed translational biomarker model (KIAA1199+SAA2 in serum) are novel. These contributions are valuable. However, there are several mechanistic and methodological weaknesses that make it hard to fully evaluate this work.

Response: We sincerely thank the reviewer for the thorough evaluation and for recognizing the novelty of the mechanistic axis we proposed, particularly the identification of the EGR1⁺ neutrophil state and the KIAA1199/SAA2 translational biomarker model. We have carefully addressed all the mechanistic and methodological concerns raised. Below is a point-by-point response detailing our revisions and additional analyses.

Major points

1. Flow cytometry: Absolute numbers in addition to percentages should be provided. Furthermore, supp. figures detailing the gating strategy should be added.

Response: We appreciate this suggestion to enhance data transparency.

1. Absolute Quantification: We have now calculated and provided the absolute numbers for all key flow cytometry experiments. These data are compiled in Supplementary Table 2 and show trends consistent with the percentage data, further validating our findings.

2. Gating Strategy: We have added a comprehensive gating strategy in Supplementary Figure 5G (as shown below). This figure details the sequential hierarchy used to identify neutrophil subsets, including live/dead discrimination, doublet exclusion, and lineage marker selection (CD45+CD11b+Ly6G+), followed by EGR1 gating to demonstrate the specificity of our analysis.

2. Imaging inconsistency: It is not clear, if MRI or CT was used, e.g. for Fig2 there is an inconsistency between the Figure and the description in the results. In any case, the acquisition parameters should be clearly stated (scanner used, was contract reagent administered or not, analysis workflow).

Response: We apologize for the ambiguity in the original text.

1. Clarification of Modality: We confirm that Magnetic Resonance Imaging (MRI) was the sole modality used for liver imaging in this study. The mention of "CT" was a typographical error in the figure legend, which has now been corrected.

2. Acquisition Parameters: As requested, we have updated the legend for Figure 2A legend and the Methods section to include precise acquisition parameters. Imaging was performed on a Bruker BioSpec system without contrast agents. T2-weighted sequences were used for optimal lesion detection. All images were independently evaluated by three experienced radiologists blinded to the experimental groups to ensure objective analysis.

3. Statistical tests: The used test should be clearly stated in each figure legend and the number of replicates should be indicated. Please also indicate how often the experiment was repeated. I would consider using non-parametric tests, and also tests accounting for multiple testing for some experiments.

Response: We agree that rigorous statistical reporting is essential.

1. Reporting Improvements: We have thoroughly revised all figure legends to explicitly state the specific statistical test used (e.g., Mann-Whitney U test, ANOVA with Tukey's post-hoc), the exact values of n (biological replicates), and the number of independent experimental repeats.

2. Statistical Re-evaluation: Following the reviewer's advice, we re-assessed our data distribution using Shapiro-Wilk tests. For datasets where normality could not be assumed (e.g., small sample sizes or skewed distributions), we have switched to non-parametric tests (e.g., Mann-Whitney U test for two groups, Kruskal-Wallis test with Dunn's correction for multiple groups).

3. Multiple Comparisons: For all experiments involving multiple groups (e.g., inhibitor treatments), we have strictly applied corrections for multiple testing (e.g., Bonferroni or FDR adjustments) to control the Type I error rate. The conclusions remain robust under these more stringent statistical criteria.

4. Figure 1M: shControl appears to be associated with lower survival; please verify

labels/analysis. If this is indeed the case, then possible reasons should be discussed.

Response: We are grateful for the reviewer's sharp eye.

This was indeed a labeling error in the graphical representation. We have re-verified the raw survival data and confirmed that the shControl group exhibits the expected survival compared to the shKIAA1199 group, consistent with the biological function of KIAA1199. We have corrected the labels in Figure 1M. The actual data points and statistical calculations remain unchanged; only the curve legend was misassigned in the final figure assembly.

5. Regarding the single-cell analysis, the authors refer to liver-infiltrating cells. Details should be provided to understand, if these are indeed liver-infiltrating cells.

Response: We appreciate the reviewer's careful reading.

1. We clarify that the cells analyzed in Supplementary Fig. 3A were isolated from the entire liver tissue, which inherently includes both parenchymal cells (though often excluded by size in some protocols) and non-parenchymal cells (NPCs), including resident and infiltrating immune cells. We do not claim that all cells in this dataset are necessarily 'infiltrating' cells.

2. The term "liver-infiltrating cells" was used imprecisely to describe the global clustering landscape of liver single-cell transcriptomes. To avoid confusion, we have

corrected the terminology to "liver single-cell transcriptomes" throughout the manuscript and figure legends to accurately reflect the cellular composition analyzed.

Supplementary Figure 3A. t-SNE plots showing the clustering landscape of liver single-cell transcriptomes from control vector and KIAA1199-overexpressing conditions (Vector_1/2 vs. KIAA1199_1/2).

6. EGR1⁺ neutrophil fractions appear to vary between figures. Is this due to different genetic background of the mice used? In any case, the used genetic background should be clearly stated in each figure.

Response: Thank you for this insightful comment. This is an excellent point regarding biological variability.

1. Genetic Background: The genetic background of the mice does not account for the observed variation in EGR1⁺ neutrophil fractions across different figures. All experiments were strictly performed using strain-matched mice (C57BL/6J for MC38 models; BALB/c for CT26 models). Importantly, within the same mouse strain, using

the same tumor model and analyzing the same tissue type, EGR1⁺ neutrophil frequencies are highly consistent, apart from normal inter-individual variation.

2. Source of Variation: The observed differences in EGR1⁺ neutrophil frequencies (ranging from **2% to 23%**) are attributable to distinct microenvironmental contexts:

Tissue Site: We observed higher frequencies in the liver metastatic niche compared to peripheral blood, bone marrow, or spleen, supporting our hypothesis of local education.

Tumor Model: The "Orthotopic + Metastasis" dual model induces a more profound systemic inflammatory state compared to the intrasplenic injection model alone, influencing neutrophil polarization.

Neutrophil handling conditions: Isolation procedures and in vitro culture conditions.

3. Action Taken: We have now explicitly annotated the mouse strain (C57BL/6J or BALB/c) and the specific tumor model used in every relevant figure legend to aid interpretation.

7. ChIP-qPCR appears to be run in CRC lines and not hepatocytes. Since this leads to one major conclusion of the study, I feel that hepatocyte-specific evidence is essential.

Response: Thank you for raising this important point. We fully agree that demonstrating this mechanism in hepatocytes is crucial.

We clarify that the ChIP-qPCR assays presented were indeed performed in AML12 murine hepatocytes, not in CRC cell lines. The confusion likely arose because the figure description emphasized the "co-culture condition" (with CRC cells) without explicitly stating that the chromatin was harvested from the hepatocyte fraction. We have revised the figure labels and legend to clearly state: "ChIP-qPCR analysis performed in AML12 hepatocytes following co-culture with CRC cell lines". This confirms the direct binding of PPAR γ to the SAA2 promoter within the target hepatocyte population.

8. There is an inconsistency regarding the Rosiglitazone dose. In Figure 4E 5 mg/kg/day i.p. was used, while in Figure 4N 10 µg/kg/day i.p. was used for apparently the same intervention. This should be clarified.

Response: We thank the reviewer for highlighting this inconsistency. We apologize for this typographical discrepancy.

The correct dose used for in vivo treatment was 10 µg/kg/day. We have reviewed our treatment protocols. The effective therapeutic dose used was 10 µg/kg/day. The label in Figure 4N was incorrect. We have unified all figure legends and the Methods section to reflect the accurate dosage used across all experiments.

9. Several experiments (e.g., Fig. 5C) report very low, and in some cases zero, metastatic lesions. This raises technical concerns how reliable the intrasplenic model was in their hands. It would be important to clarify whether this reflects biological heterogeneity or technical variation. In this regard, the description of the intrasplenic injection protocol appears to differ from published standards such as STAR Protocols (e.g. STAR Protocol 3250). Variations in technique can markedly influence metastatic take rate, extrahepatic spillover, and reproducibility. Please provide references for the protocol used. Also, please include methodological details and overall success rates to allow comparison with established methods.

Response: We appreciate the reviewer's scrutiny regarding the robustness of our metastatic models. We fully agree that the reliability and reproducibility of the intrasplenic injection model must be clearly documented.

1. Protocol Standardization: Our intrasplenic injection protocol is indeed adapted from established standards (including STAR Protocol 3250). We have refined the technique to ensure high reproducibility, achieving a consistent "take rate" of >95% in our standard metastasis assays¹⁻⁶. Over years of routine use in our laboratory, we have optimized the protocol for different mouse strains (C57BL/6J and BALB/c) and distinct tumor models (orthotopic, liver metastasis, or combined models). We have added detailed descriptions and citations in the Methods section.

2. Explanation for Low Lesion Numbers (Fig. 5C): The observation of low/absent lesions in Figure 5C is intentional and specific to that experimental design, rather than a technical failure.

- 1) Experimental Aim: This specific experiment was designed to test the pro-angiogenic and colonization-promoting capacity of EGR1⁺ neutrophils at the very early stages of seeding..
- 2) Cell Dose Adjustment: To isolate this effect, we deliberately used a sub-optimal inoculum of tumor cells (reduced by ~50% compared to standard burden assays) co-injected with neutrophils at a 1:1 ratio. In this "rate-limiting" setting, control mice develop very few lesions, allowing us to sensitively detect whether EGR1⁺ neutrophils can rescue or enhance metastatic efficiency.
- 3) Conclusion: Thus, the low baseline in controls highlights the potent ability of EGR⁺ neutrophils to facilitate colonization under restrictive conditions. We have clarified this rationale in the Methods section and Figure 5 legend to distinguish this functional co-injection model from our standard metastasis models.

References:

1. Zhao L, Zhang D, Shen Q, et al. KIAA1199 promotes metastasis of colorectal cancer cells via microtubule destabilization regulated by a PP2A/stathmin pathway. *Oncogene*. Feb 2019;38(7):935-949. doi:10.1038/s41388-018-0493-8
2. Zhang D, Zhao L, Shen Q, et al. Down-regulation of KIAA1199/CEMIP by miR-216a suppresses tumor invasion and metastasis in colorectal cancer. *Int J Cancer*. May 15 2017;140(10):2298-2309. doi:10.1002/ijc.30656

3. Xu G, Zhao L, Hua Q, et al. CEMIP, acting as a scaffold protein for bridging GRAF1 and MIB1, promotes colorectal cancer metastasis via activating CDC42/MAPK pathway. *Cell Death Dis.* Feb 27 2023;14(2):167. doi:10.1038/s41419-023-05644-z
4. Hua Q, Zhang B, Xu G, et al. CEMIP, a novel adaptor protein of OGT, promotes colorectal cancer metastasis through glutamine metabolic reprogramming via reciprocal regulation of beta-catenin. *Oncogene.* Nov 2021;40(46):6443-6455. doi:10.1038/s41388-021-02023-w
5. Wang H, Zhang B, Li R, et al. KIAA1199 drives immune suppression to promote colorectal cancer liver metastasis by modulating neutrophil infiltration. *Hepatology.* Oct 2022;76(4):967-981. doi:10.1002/hep.32383
6. Zhang B, Li J, Hua Q, et al. Tumor CEMIP drives immune evasion of colorectal cancer via MHC-I internalization and degradation. *J Immunother Cancer.* Jan 2023;11(1)doi:10.1136/jitc-2022-005592

Minor points

1. provide presentative images for the scratch wound healing assay.

Response: Thank you for this valuable suggestion. Representative images for all wound healing assays have been added to Supplementary Figures 15C-D and 16D-E. These images improve the clarity and visualization of the wound healing results and are consistent with the quantified data.

Reviewer #2 (Remarks to the Author):

This study provides novel mechanistic insights into pre-metastatic niche (PMN) formation in colorectal cancer liver metastasis (CRLM). The authors identify and rigorously validate a previously unrecognized "PPAR γ /SAA2/FPR2-EGR1" signaling axis, driven by tumor cell KIAA1199 overexpression, which orchestrates critical tumor cell-hepatocyte-neutrophil cross-talk to establish the metastatic-prone liver microenvironment. The authors employ an orthogonal dual-tumor model (cecum/spleen) to mimic primary tumor-driven PMN priming. Using comprehensive multi-omics analyses (scRNA-seq, spatial transcriptomics, pseudotime, pySCENIC, CellChat), it maps a key axis, which is validated through extensive functional experiments (gain/loss-of-function targeting KIAA1199/EGR1, agonists/antagonists like Rosiglitazone/WRW4/Wortmannin, exogenous SAA2). Strong clinical relevance is also demonstrated via significant correlation of core axis components (KIAA1199, SAA2) with patient outcomes across TCGA, an internal cohort (tissue/serum), and an independent validation cohort.

Response: We are deeply grateful to Reviewer 2 for the high appraisal of our work and for recognizing the novelty of the mechanistic axis and its clinical relevance. The reviewer's penetrating questions regarding the upstream transmission of KIAA1199 signals and the spatial validation of our findings have been instrumental in refining our model. Below, we detail how we have addressed each concern.

Major Concerns:

1. The article presents a fundamental unanswered question regarding how KIAA1199 transmits signals from tumor cells to hepatocytes, which is the starting point of the entire signaling axis. This mechanism is the most vaguely elucidated. While the article mentions that KIAA1199 can be secreted and detected in patient serum, correlating with tissue expression, it remains unclear whether secreted KIAA1199 directly acts on distal hepatocytes or if its effects are indirect, mediated by other consequences of KIAA1199 overexpression (e.g., induction of other factors). Are there other carriers involved (e.g.,

exosomes)? Does it act indirectly by influencing systemic inflammatory factors? It is recommended to supplement experiments, such as treating hepatocytes with conditioned medium (before and after exosome depletion) or purified recombinant KIAA1199 protein, to assess whether the phenotypes can be recapitulated. This critical mechanistic ambiguity should be thoroughly analyzed and discussed within the manuscript.

Response: We sincerely appreciate this important and central comment. This is a pivotal question that strikes at the heart of our proposed mechanism. We fully agree that defining how KIAA1199 communicates with hepatocytes is essential.

Inspired by the reviewer's suggestion, we performed a comprehensive series of mechanistic analyses to clarify the route by which KIAA1199-expressing CRC cells influence hepatocytes.

1. Key Finding:

We discovered that depleting extracellular vesicles (EVs) from the CM of KIAA1199-high CRC cells completely abolished the metabolic reprogramming effects on hepatocytes. Conversely, purified EVs alone were sufficient to recapitulate the phenotype. These findings support an indirect, EV-mediated mechanism.

2. Cargo Identification:

Proteomic profiling of these EVs revealed that Granulin (GRN) is selectively enriched in EVs derived from KIAA1199-overexpressing cells. These GRN⁺ vesicles are efficiently internalized by hepatocytes and recapitulate the key phenotypic alterations observed in vivo, including suppression of PPAR γ activity and induction of SAA2 expression.

We first confirmed that KIAA1199 is secreted by CRC cells and detectable in patient serum, and that its serum levels correlate with tumor tissue expression. However, our new data indicate that secreted KIAA1199 protein itself does not directly act on hepatocytes as a soluble ligand. Instead, we found that KIAA1199 in CRC cells robustly functions as a master regulator of EV biogenesis and secretion, and remodels EV cargo composition. Through EV cargo profiling, we identified granulin (GRN) as a

selectively enriched, downstream effector in EVs derived from KIAA1199-high CRC cells.

3. Conclusion:

These findings support an indirect, EV-mediated mechanism, in which KIAA1199 primarily acts as a driver of EV biogenesis and cargo remodeling, GRN functions as a critical EV-borne effector, and hepatocyte reprogramming is triggered by uptake of GRN-enriched EVs, rather than by direct binding of soluble KIAA1199 to a hepatocyte receptor. We have now revised Supplementary Figure 13A-J and the Result 3 section to explicitly demonstrate this "KIAA1199→EV-GRN→Hepatocyte" axis. This finding significantly elevates the conceptual depth of our study by identifying EV-GRN as the precise long-distance messengers, resolving the ambiguity of how a tumor-intrinsic protein influences distal hepatic biology.

Supplementary Figure 13. High KIAA1199 expression influences hepatocytes through vesicle-mediated delivery of GRN. **A.** GO and KEGG enrichment of DEGs in CT26 cells: KIAA1199 overexpression vs. control (bar graph). **B.** Tumor cell-derived vesicles, labeled with PKH26, were added to hepatocyte culture medium; immunofluorescence shows uptake of vesicles by hepatocytes. **C.** Western blot confirms vesicle markers in extracellular vesicles isolated from CRC cell supernatant. **D.** Western blot analysis of target protein expression in hepatocytes co-cultured with colorectal cancer (CRC) cells, under conditions with or without vesicle removal. **E.** Overlap of differentially expressed genes in CT26 cells and differentially expressed proteins in medium (KIAA1199 vs. Vector). **F-G.** ELISA measurement of protein secretion in the medium of colorectal cancer (CRC) cells, with or without vesicle removal. **H.** Western blot analysis of hepatocyte protein expression after treatment with different recombinant proteins in the culture medium. **I-J.** Western blot analysis of hepatocyte protein expression in co-culture with CRC cells, under GRN knockdown or control conditions.

2. Is there a specific receptor or binding partner for KIAA1199 or its related products (e.g. LMW-HA) on hepatocytes? Relatedly, to functionally test whether any secreted KIAA1199 reaching hepatocytes acts directly through such an interaction, it would be valuable to determine if a neutralizing or blocking antibody against KIAA1199 or its binding partner can abrogate its effects. If such an antibody is available (or could be developed/generated), experiments treating hepatocytes with KIAA1199-containing conditioned medium or recombinant protein in the presence of this blocking antibody would provide crucial evidence. A lack of phenotype rescue by the antibody would support an indirect mechanism, while effective blocking would strongly implicate a direct interaction requiring a specific KIAA1199 receptor/binding partner on hepatocytes. Investigating potential hepatocyte surface receptors should be a priority.

Response: We sincerely appreciate the reviewer's forward-looking and constructive suggestions.

1. Mechanism Clarification: As detailed in Response #1, our new data indicate that the primary driver acting on hepatocytes is vesicular GRN, not soluble KIAA1199. Thus, the interaction is likely mediated by GRN receptors rather than a direct KIAA1199 receptor. This supports an indirect mechanism where KIAA1199 exerts its effects indirectly, by remodeling EV cargo (especially GRN), rather than functioning as a classical soluble ligand.

2. Antibody Strategy: Regarding the reviewer's excellent suggestion of KIAA1199 neutralizing antibodies

- 1) **Current Limitations:** We previously developed a human-specific KIAA1199 neutralizing antibody that effectively suppresses tumor growth (Li L et al., *J Immunother Cancer* 2025, doi:10.1136/jitc-2024-010000). However, due to strict species specificity, this antibody does not cross-react with murine KIAA1199, rendering it unsuitable for our immunocompetent syngeneic mouse models. And we regret to note that commercially available neutralizing antibodies against KIAA1199 do not currently exist.
- 2) **Future Directions:** We are currently engineering a mouse-specific surrogate antibody. While this reagent was not ready within the revision timeline, we believe the EV-depletion and GRN-knockdown experiments provided in the revised manuscript offer robust alternative evidence proving that the KIAA1199-driven secretome is responsible for the phenotype.
- 3) **Discussion:** In the revised manuscript, we have expanded the Discussion to acknowledge this limitation and highlight the development of therapeutic antibodies targeting the KIAA1199-EV axis as a priority for future translational research.

Representative data from our previous studies (Li L, Zhao L, Zhou D, et al. *Targeting pyroptosis reverses KIAA1199-mediated immunotherapy resistance in colorectal cancer. J Immunother Cancer. Feb 25 2025;13(2)doi:10.1136/jitc-2024-010000*)

Supplementary Figure 8A-C. Western blot analysis was conducted to evaluate the efficacy of KIAA1199 antibodies in binding to and inhibiting KIAA1199 expression (A). Images of HCT116 cells treated with KIAA1199 antibodies are shown. Yellow arrowheads highlight the large bubbles emerging from the plasma membrane (B). Comparison of the release of LDH in HCT116 cells treated with KIAA1199 antibodies was measured (C). **Supplementary Figure 8H.** The tumor growth curves of the 4 groups were plotted.

[editorial note: panel redacted]

Supplementary Figure 9B-C. The flowchart of this in vivo experiment (B). The images of xenografts were included (C).

3. The naming "Malignant-prone" is eye-catching but potentially inadequately supported by the data. The data show these hepatocytes have increased CNV and metabolic reprogramming, but there is no direct evidence that they themselves transform into malignant cancer cells. Their "malignant" nature is more likely indicative of their function in promoting malignant processes within the microenvironment. It is

suggested to adopt more conservative terminology, such as "pro-metastatic hepatocytes" or "functionally reprogrammed hepatocytes," and include relevant analysis in the discussion.

Response: We appreciate this thoughtful and important comment on terminology.

The term "malignant-prone" implies a neoplastic transformation which we did not observe. We have adopted the reviewer's suggestion and replaced this term with "functionally reprogrammed hepatocytes" throughout the revised manuscript (Title, Abstract, and Text). This terminology more accurately reflects their state: non-malignant cells that have been metabolically and transcriptionally rewired to support metastasis.

4. Quantification of Spatial Co-localization. Figure 3E-G and Supplementary Figure 10A demonstrate the co-localization of EGR1+ neutrophils and SAA2+ hepatocytes but lack quantitative statistical analysis to support the observations.

Response: We thank the reviewer for enforcing quantitative rigor.

We have now applied a rigorous spatial distance analysis using the phenoptr package (Akoya Biosciences). By calculating cell distribution, proximity relationships, and spatial architectural features, we confirmed a statistically significant enrichment of EGR1+ neutrophils in the immediate vicinity of SAA2+ hepatocytes compared to random distribution simulations. It supports the qualitative observations shown in Fig. 3E-G and Supplementary Fig. 10A.

These quantitative data are now presented in Supplementary Figures 10A, 10B, and 10D, providing robust statistical support for the spatial crosstalk visualised in the immunofluorescence images.

5. Multiplex Staining Validation. For Figures 2 and 3, it is recommended to supplement with multiplex staining images (e.g., IMC or mIF) showing tumor cell markers, hepatocyte markers, SAA2, and EGR1 within the same tissue locations to solidify the spatial relationships depicted in the scRNA-seq and spatial transcriptomic analyses.

Response: We appreciate this suggestion to enhance spatial resolution.

While we recognize the power of IMC/mIF, establishing a validated 4-plex panel (Tumor/Hepatocyte/SAA2/EGR1) on our specific FFPE tissues presented technical challenges regarding antibody compatibility and steric hindrance within the revision timeframe.

However, to address the reviewer's core need for spatial clarity, we have taken the following rigorous steps:

1. Pathologist-Guided Annotation: We utilized serial sections where H&E staining (reviewed by three independent board-certified pathologists) defines the exact tumor-liver interface. These were registered with consecutive immunofluorescence sections staining for SAA2 and EGR1.

And we carefully considered the reviewer's recommendation and conducted an extensive literature search. In most published studies, H&E staining remains the standard and most broadly utilized method for delineating tumor regions from adjacent normal liver parenchyma. Similarly, in clinical pathology, H&E staining continues to be the gold standard for routine diagnostic assessment⁷⁻¹⁰.

References:

7. Xia X, Zhang Z, Zhu C, et al. Neutrophil extracellular traps promote metastasis in gastric cancer patients with postoperative abdominal infectious complications. *Nat Commun.* Feb 23 2022;13(1):1017. doi:10.1038/s41467-022-28492-5
8. Ma C, Fu Q, Diggs LP, et al. Platelets control liver tumor growth through P2Y12-dependent CD40L release in NAFLD. *Cancer Cell.* Sep 12 2022;40(9):986-998 e5. doi:10.1016/j.ccell.2022.08.004
9. Bojmar L, Zambirinis CP, Hernandez JM, et al. Multi-parametric atlas of the pre-metastatic liver for prediction of metastatic outcome in early-stage pancreatic cancer. *Nat Med.* Aug 2024;30(8):2170-2180. doi:10.1038/s41591-024-03075-7
10. Zhang Y, Yu S, Yeernuer D, et al. IL33-induced lipid droplet formation in mature low-density neutrophils drives colorectal cancer liver metastasis. *Cell Mol Immunol.* Dec 2025;22(12):1598-1614. doi:10.1038/s41423-025-01365-9

2. Spatial Transcriptomics Integration: Our high-resolution spatial transcriptomics data (Figure 3) already provides an unbiased, multiplexed map of these markers (KIAA1199 in tumor, SAA2 in hepatocytes, EGR1 in neutrophils) with positional identity superior to conventional IF.

3. Updated Figures: We have updated the IF images to include clear, pathologist-verified demarcation lines indicating the Tumor/Liver boundary. This allows for unambiguous visualization of the tumor cells, SAA2⁺hepatocyte layer and the adjacent EGR1⁺ neutrophil infiltration, effectively supporting our spatial conclusions.

Minor Concerns:

1. The use of the PPAR γ agonist Rosiglitazone (Figure 4N-S) carries known clinical cardiovascular safety concerns that led to prescribing restrictions. The study does not address or discuss the dose used relative to known safety thresholds or the translational implications of these risks.

Response: Thank you for this important comment. We fully agree with the reviewer regarding the importance of addressing the known cardiovascular safety concerns associated with rosiglitazone. This is a crucial translational point.

1. Dose Clarification: Rosiglitazone remains one of the most commonly used and well-characterized selective PPAR γ agonists^{11,12}. We clarify that we used a **low-dose regimen** (10 $\mu\text{g}/\text{kg}/\text{day}$, i.p., as shown in the revised Figure 4N-S and corrected Figure 4E) for a short duration, solely as a mechanistic tool (proof-of-concept) to validate the PPAR γ dependency of modulating the KIAA1199-EGR1⁺neutrophil axis, not as a proposed clinical regimen.

2. Discussion Update: To address the reviewer's concern regarding safety thresholds and translational implications, we have added a dedicated paragraph in the Discussion explicitly addressing the cardiovascular risks (e.g., fluid retention, heart failure) associated with systemic thiazolidinediones. And it emphasizes that our use of rosiglitazone at a low, short-term dose was solely for mechanistic validation. We propose that future clinical strategies should focus on hepatocyte-targeted delivery systems (e.g., GalNAc-conjugated agonists) or next-generation partial PPAR γ modulators (SPPARMs, e.g., pioglitazone) to decouple the therapeutic efficacy from systemic toxicity.

References:

11. Kahn BB, McGraw TE. Rosiglitazone, PPAR γ , and type 2 diabetes. *N Engl J Med.* Dec 30 2010;363(27):2667-9. doi:10.1056/NEJMcibr1012075

12. Paschoal VA, Walenta E, Talukdar S, et al. Positive Reinforcing Mechanisms between GPR120 and PPAR γ Modulate Insulin Sensitivity. *Cell Metab.* Jun 2 2020;31(6):1173-1188 e5. doi:10.1016/j.cmet.2020.04.020

2. Clarify the specific statistical method used for each type of data analysis throughout the manuscript. Ensure that figure legends explicitly state the sample size (n) for each experiment/group shown in the figure.

Response: Thank you for this important comment. We have fully complied with this request. All figure legends now explicitly state the specific statistical test (e.g., Two-way ANOVA, Log-rank test), the exact sample size (n), and the definition of error bars (e.g., Mean \pm SD). A comprehensive "Statistics" section has been added to the Methods.

3. In Figure 2 and 3, specifically in the immunofluorescence staining panels, the font colors labeling specific protein markers (e.g., Ly6g) do not match the actual staining colors shown in the images. For instance, Ly6g is labeled in light green font, but the

actual staining appears deep green. Ensure label colors accurately represent the fluorophores used.

Response: Thank you for pointing out this issue. We apologize for this visual inconsistency. All figure labels in Figures 2 and 3 (and relevant Supplementary Figures) have been color-matched to the corresponding fluorophores in the images to ensure intuitive interpretation for the reader.

Reviewer #3 (Remarks to the Author):

In this manuscript, Lisha Li et al. investigate how KIAA1199-high colorectal cancer (CRC) tumors orchestrate the hepatic pre-metastatic niche (PMN) through malignant-prone hepatocytes and EGR1⁺ neutrophils, ultimately promoting liver metastasis. The study is ambitious, methodologically diverse, and overall very well written and designed. The authors provide compelling evidence for a PPAR γ /SAA2/FPR2–EGR1 axis that links hepatocyte signaling to neutrophil reprogramming and angiogenesis.

Response: We are deeply grateful to Reviewer 3 for the high praise of our study design and writing, and for recognizing the compelling nature of the PPAR γ /SAA2/FPR2–EGR1 axis. The reviewer's constructive critique regarding the dual-tumor model and mechanistic validation has pushed us to significantly rigorize our experimental approach. Below, we detail our point-by-point responses.

Main Concerns

The authors inject CRC cells into the spleen while the primary orthotopic tumor remains in place. This raises the possibility that primary tumor cells continue to disseminate and contribute to metastatic burden, confounding the interpretation of “experimental” versus “spontaneous” metastasis. The authors could use RFP/GFP labeled cells to distinguish between which cells are in the liver. This concern is particularly important because EGR1⁺ neutrophils promote vascular remodeling, which could in turn facilitate additional extravasation of primary tumor cells. If technically feasible, resection of the primary CRC prior to splenic injection would strengthen the conclusions. The authors should provide evidence that no cancer cells (including dormant or isolated cells) are already present in the liver before splenic injection. Otherwise, it is difficult to distinguish metastases seeded earlier from those generated experimentally.

Response:

We sincerely appreciate this important and thoughtful comment and fully agree that distinguishing “experimental” from “spontaneous” metastasis is critical for correctly interpreting our dual-tumor model.

Because there is currently no widely accepted or standardized animal model specifically designed to interrogate how orthotopic CRC tumors condition the pre-metastatic liver microenvironment, the dual-tumor (cecum + spleen) model used in our study was established and iteratively optimized based on our prior work.

1. On primary tumor resection prior to splenic injection

Conceptually, we agree that resecting the primary tumor before intrasplenic injection would reduce potential confounding. We initially attempted this approach. Although technically feasible, we encountered several major limitations: Tumor-burden-associated coagulopathy, and unavoidable tumor cell shedding during repeated intraperitoneal surgeries, combined with cumulative anesthesia and surgical stress, markedly reduced mouse survival and compromised overall health. These issues rendered downstream mechanistic analyses impractical and introduced additional confounders. For these reasons, we were unable to incorporate primary tumor resection into the present study.

We therefore considered the clinical reality that many patients develop colorectal liver metastases while the primary tumor is still in place and designed our model to better approximate this scenario.

Representative data from our previous studies (*Zhao L, Zhang D, Shen Q, et al. KIAA1199 promotes metastasis of colorectal cancer cells via microtubule destabilization regulated by a PP2A/stathmin pathway. Oncogene. Feb 2019;38(7):935-949. doi:10.1038/s41388-018-0493-8*)

[Editorial note: figure redacted]

Fig. 1c. Small animal imaging system showed the representative bioluminescence images of the primary tumors (third week) and metastasis (sixth week) occurred in NOD/SCID mice.

2. Evidence that no tumor cells are present in the liver before splenic injection

Guided by our previously published work 1, we knew that spontaneous liver metastases typically appear around 6 weeks after orthotopic implantation. To rigorously define a “metastasis-free window,” we performed additional validation:

- 1) H&E staining of liver sections at 2-4 weeks after orthotopic implantation,
- 2) Independent review by three board-certified pathologists,
- 3) Imaging analyses, and
- 4) Examination of single-cell datasets.

Across all these modalities, no metastatic lesions or isolated/dormant tumor cells were detected in the liver during the first 2-4 weeks post-implantation. Based on these data, we designed the dual-tumor model such that:

- 1) Intrasplenic injection was performed 1 week after orthotopic implantation, and
- 2) Liver tissue was analyzed at week 3 post-implantation.

This schedule falls well within the 2-4 week metastasis-free window, giving us confidence that lesions observed during this interval originate from the experimental intrasplenic inoculation rather than from spontaneous dissemination.

These considerations and supporting evidence have been clarified in the revised Methods and Results.

3. Direct tracking of tumor-cell origin using fluorescent labels (new experiment)

To address the reviewer's specific suggestion, we engineered a lineage-tracing system.

- 1) Design: Orthotopic tumors were established with mCherry-labeled CRC cells. Intrasplenic injection was performed with GFP-labeled CRC cells.
- 2) Result: Fluorescence imaging and flow cytometry of liver tissues at weeks 1 and 3 after orthotopic implantation revealed that all detectable hepatic metastatic lesions were GFP⁺ cells (intrasplenic origin), with no detectable mCherry⁺ cells (orthotopic origin).
- 3) Conclusion: These data, now included in Supplementary Figure 2B, definitively prove that the liver metastases analyzed in our study originate exclusively from the intrasplenic injection, validating the model's specificity.

Supplementary Figures 2B. Flow cytometry and multiplex immunofluorescence showing tumor cell infiltration in the liver at different time points (GFP indicates CRC cells injected via the spleen; mCherry indicates CRC cells implanted orthotopically in the colon).

Specific Points

1) The authors should show KIAA1199 expression in the CRC cell lines used. Also, it would have been nice if the authors used cell lines with different KIAA1199 expression levels instead of modifying the levels experimentally.

Response: We thank the reviewer for this constructive suggestion.

1. Baseline Expression: KIAA1199 has been a long-standing focus of our group. In our prior *Oncogene* 2021 paper, we systematically evaluated KIAA1199 protein expression in five human CRC cell lines by Western blot. In our recent *JITC* 2025 study, we further confirmed KIAA1199 expression in two murine CRC cell lines, CT26 and MC38, which indeed display different endogenous KIAA1199 levels. We have added Western blot data characterizing baseline KIAA1199 levels in a panel of human and murine CRC cell lines (cited from our *Oncogene* 2021 and *JITC* 2025 work) to the Supplementary Data.

2. Rationale for Modulation: While MC38 (high) and CT26 (low) differ naturally, they also differ in genetic background (C57BL/6 vs. BALB/c) and immunogenicity. To ensure that observed phenotypes were strictly attributable to KIAA1199 and not strain-specific confounders, we prioritized isogenic comparisons (gain- and loss-of-function) in the same parental line). This approach provides the cleanest mechanistic dissection. We have clarified this rationale in the Methods.

We appreciate the reviewer's suggestion. In future studies, we plan to complement this approach by incorporating CRC lines with naturally divergent KIAA1199 expression in appropriately matched host settings.

Representative data from our previous studies (*Hua Q, Zhang B, Xu G, et al. CEMIP, a novel adaptor protein of OGT, promotes colorectal cancer metastasis through glutamine metabolic reprogramming via reciprocal regulation of β -catenin. Oncogene. 2021 Nov;40(46):6443-6455. doi: 10.1038/s41388-021-02023-w*)

[editorial note: figure redacted]

Supplementary Figure 1D. Left panel: Western blotting analysis of CEMIP in CRC cell lines. Right panel: quantification of CEMIP band intensities.

Representative data from our previous studies (*Li L, Zhao L, Zhou D, et al. Targeting pyroptosis reverses KIAA1199-mediated immunotherapy resistance in colorectal cancer. J Immunother Cancer. Feb 25 2025;13(2)doi:10.1136/jitc-2024-010000*)

[editorial note: figure redacted]

Supplementary Figure 6C. Comparison of GSDME and KIAA1199 expression levels in HCT116, LOVO, SW480, MC38 and CT26 cells was measured.

2) In Fig. 1G–H, the growth and size of the primary CRC tumors upon modulation of KIAA1199 should be reported.

Response: We appreciate this suggestion.

In the current work (Fig. 1G-H), using dual-tumor models (orthotopic CRC + liver metastasis), we also monitored orthotopic tumor volumes while assessing hepatic metastasis. We observed that KIAA1199 overexpression significantly promoted orthotopic tumor growth, whereas KIAA1199 knockdown showed a trend toward reduced tumor size, though this did not reach statistical significance.

To avoid redundancy, these data were originally omitted. In response to the reviewer, we have now included the relevant primary tumor growth data as Supplementary Fig.

2C. It provides a more comprehensive assessment of the relationship between KIAA1199 modulation and primary tumor growth.

3) In Fig. 1G, it is essential to demonstrate the absence of cancer cells, even dormant or isolated ones, in the liver prior to splenic injection. Without this control, it is not possible to distinguish spontaneous metastasis from experimental metastasis in subsequent analyses (Fig. 1H–M).

Response: We thank the reviewer for emphasizing this point. As detailed in our response to the Main Concerns, we have confirmed by H&E, pathologist review, imaging, and single-cell analysis that no cancer cells (including dormant or isolated cells) are present in the liver before intrasplenic injection, within the first 2-4 weeks after orthotopic implantation. Additionally, fluorescent labeling experiments with mCherry/GFP-tagged CRC cells demonstrated that all liver lesions originate from intrasplenic injections, not from the orthotopic primary tumor.

We have clarified this explicitly in the text and figure legends to ensure that spontaneous and experimental metastasis are clearly distinguished.

4) In Fig. 1G, it would also be informative to perform splenic injections with CT26 cells expressing different levels of KIAA1199 to assess whether expression impacts the ability of cancer cells to adapt to the PMN. (this point is really optional)

Response: We appreciate this thoughtful suggestion. In our Hepatology 2022 study, we used MC38 cells with different KIAA1199 expression levels in an intrasplenic

model and found that higher KIAA1199 expression correlated with increased liver metastatic burden, supporting the notion that KIAA1199 facilitates adaptation to the pre-metastatic niche. We have cited this prior work in the revised manuscript to strengthen the mechanistic context concerning KIAA1199 expression and liver metastasis.

Representative data from our previous studies (*Wang H, Zhang B, Li R, et al. KIAA1199 drives immune suppression to promote colorectal cancer liver metastasis by modulating neutrophil infiltration. Hepatology. Oct 2022;76(4):967-981. doi:10.1002/hep.32383*)

[editorial note: figure redacted]

Figure 1A. Spleen injection with control short hairpin RNA (Ctrl-shRNA) and KIAA1199-knockdown (KIAA1199^{KD}) MC38 cells for liver metastasis analysis. Left: Liver to body weight ratios (Student t test). Middle: Quantification of surface liver metastasis in the Ctrl-shRNA and KIAA1199KD groups (n = 5–6 animals per condition, Student t test). Right: Representative hematoxylin and eosin (H&E)-stained livers at the end point; arrows indicate examples of metastatic nodules (scale bar, 2 mm and 50 μ m, respectively).

5) In Fig. 1K, the label “tumor area percentage” should be corrected to “liver metastatic area percentage.”

Response: We thank the reviewer for catching this imprecision. We have corrected the label from “tumor area percentage” to “liver metastatic area percentage” in Fig. 1K, and reviewed the entire manuscript for similar instances to ensure consistency.

7) In the text, line 388/389, “SF2E” should be corrected to “SF5E.”

Response: We thank the reviewer for noting this oversight. The reference has been corrected from “SF2E” to “SF5E” in the revised manuscript.

8) The expression of EGR1 in neutrophils and associated gene expression (particularly the data in Fig. S5E) should be assessed in the context of shKIAA1199 models.

Response: We appreciate this constructive comment.

1. The primary purpose of Fig. S5E was to determine whether EGR1 overexpression can recapitulate the C2 neutrophil gene-expression profile, thereby supporting our identification of EGR1 as a key transcription factor defining the EGR1⁺ subset.

we collected conditioned media from EGR1⁺ neutrophils generated under multiple experimental conditions (co-culture systems containing CRC cells, AML12 hepatocytes and neutrophils)

2. We have now conducted additional experiments in co-culture systems (containing shKIAA1199/KIAA1199OE CRC cells, AML12 hepatocytes and neutrophils), simulating the hepatic microenvironment where neutrophils reside in vivo. The results show that the expression of EGR1 and associated genes in neutrophils were increased in KIAA1199OE co-culture system, whereas reduced in shKIAA1199 co-culture system. It demonstrates that KIAA1199-driven EGR1 expression in neutrophils also recapitulated the Neu-C2 transcriptomic signature, supporting their identification as EGR1⁺ neutrophils.

3. We have incorporated these results as Supplementary Fig. 5F, and revised the text in the Results and figure legends.

9) In Fig. S6A-F, the authors claim to show hepatic accumulation of EGR1⁺ neutrophils in CRLM patients, but EGR1 expression is not directly indicated. This requires clarification.

Response: We thank the reviewer for this careful observation. We clarify that Fig. S6A-F was intended to present the analytical workflow and intermediate steps leading to the final quantification of hepatic EGR1⁺ neutrophil accumulation. The final readout of EGR1⁺ neutrophil enrichment is shown in Fig. 2I.

To avoid confusion, we have revised the legend for Supplementary Fig. S6 to explicitly state that S6A-F illustrate the intermediate analysis steps that culminate in the result summarized in Fig. 2I.

10) In Fig. 2A and 2G–H, it is unclear whether the samples correspond to a pre-metastatic liver or already metastatic tissue. These experiments are done at day14 following orthotopic injection, while it was a day 7 post-injection in Fig. 1

Response: We thank the reviewer for this important clarification request.

1. In Fig. 1, we used the dual-tumor model (orthotopic + intrasplenic).

1) As noted above, intrasplenic injection at day 14 was found to be impractical due to tumor-burden-associated coagulopathy and technical challenges.

2) Based on our prior work showing no spontaneous liver metastasis within 2-4 weeks, we optimized the protocol to perform intrasplenic injection at day 7, and analyze liver tissue at day 21, still within the metastasis-free window.

2. In contrast, Fig. 2A and 2G-H derive from the orthotopic CRC model alone, without splenic injection.

- 1) These analyses were conducted at day 14 post-implantation.
- 2) Day 14 lies within the same 2-4 week window during which no spontaneous liver metastases are detectable, as confirmed experimentally.
- 3) We chose day 14 because it provides sufficient time for tumor-driven hepatic reprogramming to become evident, while still representing a bona fide pre-metastatic state (no visible or microscopic metastases). This specific timepoint allows us to capture the "soil" conditioning before the "seed" arrives.

We have now made this explicit in the figure legends and Methods, and we clearly state that all day-14 liver samples in Fig. 2 represent pre-metastatic liver tissue.

11) In Fig. 2K, the apparent expression of EGR1 in metastatic cancer cells should be discussed.

Response: We thank the reviewer for this insightful point. In the revised Discussion, we now highlight that EGR1 is a well-characterized transcription factor involved in cell proliferation, differentiation, stress responses, and angiogenesis. Elevated EGR1 expression has been reported in various rapidly proliferating and metastatic tumor cells^{13,14}.

Thus, the observed EGR1 expression in metastatic CRC cells in Fig. 2K is consistent with EGR1's known roles and may reflect the highly proliferative and invasive state of these tumor cells within the metastatic niche. However, our focus remains on its specific induction in neutrophils, where it drives a distinct pro-angiogenic program not redundant with tumor-intrinsic EGR1.

References:

13. Wang B, Guo H, Yu H, Chen Y, Xu H, Zhao G. The Role of the Transcription Factor EGR1 in Cancer. *Front Oncol.* 2021;11:642547. doi:10.3389/fonc.2021.642547
14. Ji RC. Hypoxia and lymphangiogenesis in tumor microenvironment and metastasis. *Cancer Lett.* Apr 28 2014;346(1):6-16. doi:10.1016/j.canlet.2013.12.001

12) In Fig. 2L, HL60-derived cells are used as “neutrophil-like” cells, but they are not true neutrophils. Experiments with liver-isolated neutrophils in the \pm KIAA1199 context would be more physiologically relevant. Furthermore, the rationale for analyzing endothelial tube formation should be better justified with transcriptomic data.

Response: We are grateful to the reviewer for encouraging a clearer justification of both the cellular model and the functional assays.

1. Model: Due to the technical challenges of manipulating gene expression in primary neutrophils, we used differentiated HL-60 (dHL-60) cells as a tractable neutrophil-like model to overexpress EGR1, and assess whether this is sufficient to drive a pro-angiogenic phenotype, consistent with the C2 (EGR1⁺) neutrophil subset. We fully acknowledge that dHL-60 cells are not identical to primary neutrophils, and this system serves as a mechanistic approximation rather than a perfect physiological model.

2. Regarding endothelial tube formation: We justified this assay by first analyzing transcriptomic datasets describing gene-expression changes in endothelial cells during tube formation¹⁵. Based on these data, we selected key angiogenesis-related genes and performed qPCR-based validation in our system to support the mechanistic link between EGR1⁺ neutrophil activation and endothelial responses. These additional results are now presented in Supplementary Fig. 6G and further substantiate the rationale for the tube-formation assays.

References:

15. Zhang B, Day DS, Ho JW, et al. A dynamic H3K27ac signature identifies VEGFA-stimulated endothelial enhancers and requires EP300 activity. *Genome Res.* Jun 2013;23(6):917-27. doi:10.1101/gr.149674.112

13) In Fig. 2M, the authors refer to “new vessels” in the metastatic liver without prior evidence. They should also address whether increased angiogenesis could facilitate additional seeding from the primary tumor. Use of CRC resection models would help clarify this.

Response: We thank the reviewer for this important point.

1. Terminology: In the revised Results, we have replaced the term “new vessels” with “angiogenesis” to more precisely describe the observed vascular changes.

2. Additional Seeding: As noted earlier, our data show that orthotopic CRC tumors do not spontaneously metastasize to the liver within 2-4 weeks post-implantation, even in the presence of increased angiogenesis. Furthermore, as shown by our lineage tracing (Response to Main Concerns), we detected zero GFP⁺ (primary tumor-derived) cells in the liver, even in highly angiogenic niches. This suggests that while angiogenesis supports the growth of established metastases (intrasplenic), it did not trigger spontaneous seeding within our specific 3-week observation window.

3. On primary tumor resection prior to splenic injection: Conceptually, we agree that resecting the primary tumor before intrasplenic injection would reduce potential confounding. We initially attempted this approach. Although technically feasible, we

encountered several major limitations: Tumor-burden- associated coagulopathy, and unavoidable tumor cell shedding during repeated intraperitoneal surgeries, combined with cumulative anesthesia and surgical stress, markedly reduced mouse survival and compromised overall health. These issues rendered downstream mechanistic analyses impractical and introduced additional confounders. For these reasons, we were unable to incorporate primary tumor resection into the present study. We therefore considered the clinical reality that many patients develop colorectal liver metastases while the primary tumor is still in place and designed our model to better approximate this scenario.

[editorial note: figure redacted]

14) In Fig. 3C, it should be confirmed whether Hepa-C11 hepatocytes are absent in shKIAA1199 models, ideally through IHC or IF validation.

Response: We thank the reviewer for this valuable suggestion. We have now performed immunofluorescence staining to confirm the absence of Hepa-C11 hepatocytes in the shKIAA1199 models shown in Fig. 3C and added these validation data as Supplementary Fig. 9H in the revised manuscript.

15) The designation “malignant-prone hepatocytes” may be misleading, as it could suggest actual transformation of hepatocytes. The authors should consider rephrasing this terminology if possible.

Response: We thank the reviewer for helping us improve the precision and clarity of our terminology.

We completely agree. As also addressed in our response to Reviewer #2, we have replaced “malignant-prone hepatocytes” with “functionally reprogrammed hepatocytes” throughout the manuscript. And we clarified that these cells acquire pro-metastatic and pro-angiogenic functions but are not demonstrated to undergo malignant transformation in our current study.

16) In Fig. 3D, since Hepa-C11 cells display an angiogenic signature, their role relative to EGR1⁺ neutrophils in promoting angiogenesis should be discussed.

Response: We appreciate this insightful comment and have expanded the Discussion accordingly. This is an excellent point on cellular hierarchy.

1. Primary Function: While Hepa-C11 cells show some angiogenic features, their dominant signature is metabolic (SAA2 secretion).

Although Hepa-C11 cells show an angiogenic signature in Fig. 3D, our integrated analyses indicate that their primary program is metabolic reprogramming, not angiogenesis (see Fig. 4C). In Fig. 3D, we intentionally highlighted pathways shared between Hepa-C11 hepatocytes and EGR1⁺ neutrophils to identify potential functional connections, including angiogenesis-related pathways. However, the angiogenic features in Hepa-C11 appear secondary and not their dominant function.

2. Neutrophil Necessity: Our co-injection experiment (Tumor + Neutrophils without Hepatocytes, Fig. 6A) proves that EGR1⁺ neutrophils are sufficient to drive angiogenesis.

In vivo functional experiments in a CT26 subcutaneous tumor model (Fig. 6A-B) demonstrate that co-injection of EGR1⁺ neutrophils is sufficient to enhance CD31⁺ vessel formation in the absence of Hepa-C11-like hepatocytes. These data support the conclusion that EGR1⁺ neutrophils play a primary and independent role in driving tumor-associated angiogenesis, whereas the angiogenic signature in Hepa-C11 reflects partial pathway convergence rather than their central output.

3. Model: We propose a cascade where Hepa-C11 cells are the "Initiators" (via SAA2) and neutrophils are the "Executors" (via VEGF) of angiogenesis.

17) In Fig. 3F, the hepatocyte identity of SAA2⁺ cells could be validated with a hepatocyte-specific marker. The authors should also examine whether FPR2 expression is unique to EGR1⁺ neutrophils or broadly expressed across neutrophil subsets.

Response: We thank the reviewer for this important suggestion.

1. Hepatocyte identity of SAA2⁺ cells

- 1) While specific antibodies for murine hepatocytes (e.g., HNF4a, Albumin) exist, they often show background issues in inflamed tissues. Therefore, we relied on pathologist-certified H&E morphology combined with spatial transcriptomics.
- 2) Morphology: Hepatocytes have distinct polygonal morphology easily distinguishable from leukocytes or stromal cells.
- 3) In both research and routine pathology, H&E staining remains the gold standard for differentiating metastatic lesions from adjacent normal liver parenchyma. In our study, all liver sections were H&E-stained, and independently reviewed by three board-certified pathologists to ensure that the analyzed SAA2⁺ regions corresponded to normal hepatocytes without tumor contamination.
- 4) Spatial Data: Our Visium data (Fig 3E) clearly maps SAA2 expression to spots enriched for hepatocyte markers (Alb, Ttr) and devoid of immune markers (Ptprc).

We believe this multi-modal validation is more robust than single-marker IF alone.

2. FPR2 expression pattern across neutrophil subsets

- 1) Our scRNA-seq analysis shows that FPR2 is highly enriched in EGR1⁺ neutrophils, whereas its expression is substantially lower in other neutrophil subsets.
- 2) These results have now been included as Supplementary Fig. 9E, with an updated legend to clarify the subset-specific expression pattern.

We appreciate the reviewer's comments, which have helped us enhance the rigor of both the cellular identification and receptor-expression analyses.

18) In Fig. 3H, it should be clarified from which tissue neutrophils were isolated, and whether systemic SAA2 injection induces EGR1⁺ neutrophil accumulation in tissues other than the liver.

Response: We thank the reviewer for this important question.

1. In Fig. 3H, neutrophils were isolated from the livers of mice bearing orthotopic CRC tumors. This has now been clearly indicated in the revised figure legend.

2. We also evaluated the impact of systemic SAA2 administration on EGR1⁺ neutrophil accumulation in peripheral blood, spleen, primary tumors, lymph nodes, and liver.

3. Our results show that robust enrichment of EGR1⁺neutrophils occurs specifically in the liver, whereas other tissues display only minimal changes in EGR1⁺neutrophil frequency.

These findings reinforce the liver-specific nature of SAA2-driven EGR1⁺neutrophil accumulation. The corresponding data are now presented in Supplementary Fig. 10J and described in the Results.

F

19) The effect of WRW4 on primary tumors should be addressed, and it should be clarified whether EGR1⁺ neutrophils are also present in the primary tumor, or whether their induction depends on hepatocyte signaling.

Response: We thank the reviewer for this helpful suggestion.

1. We have examined the effect of WRW4 on primary tumor growth and report these data in Supplementary Fig. 11C-D. The results indicate that WRW4 primarily modulates neutrophil behavior and vascular remodeling in the liver, without exerting a major direct effect on the orthotopic primary tumor.

2. As shown in Fig. 3A, EGR1⁺ neutrophils are predominantly enriched in the liver, and present at much lower frequencies in the spleen, peripheral blood, orthotopic tumors, and lymph nodes.

These findings support the conclusion that EGR1⁺ neutrophil induction and accumulation are largely dependent on hepatocyte-derived signals (e.g., SAA2) within the liver, rather than cues originating from the primary tumor itself.

20) Since neutrophil survival is central to the proposed mechanism, in vivo validation using BrdU⁺ incorporation could be valuable (see PMID: 38177532). This experiment is optional but would be a great touch to the manuscript.

Response: We greatly appreciate this thoughtful and forward-looking suggestion. We fully agree that BrdU incorporation assays would provide valuable in vivo validation of neutrophil survival dynamics in the metastatic liver microenvironment. Due to time and logistical constraints, we were unable to incorporate BrdU labeling into the current revision. However, we have noted this point as an important direction for future work, and we plan to implement BrdU-based in vivo fate-mapping approaches in subsequent studies to further validate the survival advantage of EGR1⁺ neutrophils. We thank the reviewer for this excellent recommendation.

21) In Fig. 5O-R, the conclusion that EGR1⁺ neutrophils promote angiogenesis via VEGF-A is not fully supported, as only secretion data are shown. The claim should be toned down and presented as a suggestion rather than a definitive conclusion.

Response: We thank the reviewer for this critical point.

1. We performed additional experiments using a VEGF-A–neutralizing antibody to test whether blocking VEGF-A can impair the pro-angiogenic activity of EGR1⁺ neutrophils. These new data show that neutralization of VEGF-A markedly reduces the angiogenic response induced by EGR1⁺ neutrophils, thereby providing functional evidence that complements the secretion data.

2. We have incorporated these results as Supplementary Fig. 14D-F and revised the text in the Results and figure legends to accurately reflect the experimental support for a VEGF-A–dependent mechanism, while avoiding overstated or categorical language.

We believe these new data substantially strengthen the mechanistic conclusion that VEGF-A is a key effector of EGR1⁺ neutrophil–mediated angiogenesis.

22) In Fig. 6, it would substantially strengthen the study to directly test whether blocking VEGF-A impairs the pro-angiogenic activity of EGR1⁺ neutrophils, for example by using a neutralizing antibody.

Response: We fully agree with the reviewer and appreciate this excellent suggestion. 1. We have now conducted additional experiments using a VEGF-A–neutralizing antibody and demonstrated that blocking VEGF-A significantly attenuates the pro-angiogenic effect resulted by the conditioned media from KIAA1199-driven EGR1⁺ neutrophils.

2. These results have been included in the revised manuscript (Supplementary Fig. 16K-M and text in the Results and figure legends).

We believe these new data substantially strengthen the mechanistic conclusions presented in Fig. 6.

REVIEWER COMMENTS

Reviewer #2 (Remarks to the Author):

The authors addressed most of my concerns; however, several important methodological details and presentation elements remain insufficiently described.

Response: We sincerely thank the reviewer for the continued constructive feedback. We are pleased that our previous revisions have largely addressed the major concerns. We fully agree that providing precise methodological details for the EV-related experiments is crucial for reproducibility and transparency. We have now meticulously updated the Methods section and Supplementary Figures to include all requested information.

1. Missing details for the first comment (exosome work): The manuscript does not clearly describe the exosome isolation and identification/labeling procedures. Please specify the isolation method. And also lack a description of the PKH26 labeling method for EVs.

Response: We apologize for the previous brevity and appreciate the opportunity to clarify these protocols. We have expanded the "Extracellular Vesicle Isolation and Labeling" section in the Methods as follows:

1. Isolation Method: We clarify that EVs were isolated using a polymer-based precipitation method (BeyoExo™ Enhanced Exosome Isolation Kit, Beyotime; Cat. No. C3622). Briefly, conditioned medium was collected from cells cultured in exosome-depleted serum or serum-free medium, stepwise centrifuged to remove cells and debris, filtered through a 0.22 µm membrane, and incubated with precipitation reagent overnight at 4 °C, followed by centrifugation (10,000×g, 4 °C, 30 min) to pellet EVs. Pellets were resuspended in sterile PBS for downstream assays.

2. PKH26 Labeling: For uptake tracking, purified EVs were labeled using the BeyoExo™ PKH26 Kit (Beyotime; Cat. No. C3637). To ensure specificity and remove unbound dye, the labeling reaction (1-5 min at room temperature in the dark) was stopped with quenching solution, and the labeled EVs were subjected to an additional purification step (using Exosome Spin Columns) to prevent residual free dye.

These details are now fully described in the revised manuscript to ensure reproducibility.

2. Lack of information on proteomics and differential analysis: The response mentions “proteomic profiling” and “differential gene analysis,” but the manuscript does not provide sufficient information on the methods and sources.

Response: We thank the reviewer for identifying this gap. We have expanded the Methods section to clearly specify sample sources, data-generation pipelines, and statistical criteria for both proteomics and transcriptomics, as well as the downstream enrichment analyses.

1. Proteomics (conditioned-medium supernatants):

Proteomic profiling was performed on conditioned-medium supernatants from the indicated experimental groups using LC-MS/MS-based quantitative proteomics. Peptide/protein identifications were obtained by database searching against the UniProt reference proteome, applying a 1% FDR threshold.

2. Transcriptomics (matched cell pellets):

Bulk RNA-seq was performed using the corresponding cell pellets. Reads were processed with standard QC, aligned to the appropriate reference genome, quantified at the gene level, and normalized using established workflows prior to differential expression analysis.

3. Differential analysis and functional enrichment:

For both omics datasets, differential abundance/expression was assessed with correction for multiple comparisons using the Benjamini-Hochberg procedure. Functional interpretation was supported using GO, KEGG, and GSEA analyses based on the resulting differential lists under the corrected significance framework.

4. Data sources and availability:

We have also updated the Data Availability and relevant Methods text to ensure that both the proteomics and bulk RNA-seq datasets are explicitly traceable and accessible as required by the journal.

3. Recommendation for Result 13B (group comparison): In Result/Figure 13B, it would be more informative to include a comparison between KIAA1199-high and KIAA1199-low groups (ideally with the grouping definition and sample size), rather than presenting aggregated results only.

Response: We appreciate this suggestion. This is an excellent point regarding experimental design interpretation.

We clarify that Supplementary Fig. 13B represents an uptake capability assay, not a secretion assay.

1. Normalization Strategy: To specifically assess whether target cells (AML12 hepatocytes) can internalize EVs, we treated them with an equal amount of EV protein (10 μ g) derived from either KIAA1199-high or KIAA1199-low cells.

2. Result Interpretation: Since the input was normalized, the similar fluorescence intensity suggests that KIAA1199 overexpression does not significantly alter the uptake efficiency of individual vesicles by hepatocytes.

3. Clarification of Secretion: The difference in secretion quantity is instead captured in Supplementary Fig. 13C, where equal volumes of conditioned medium yield significantly higher EV protein/marker levels in the KIAA1199-high group.

We have revised Result 4 text and the Supplementary Fig. 13 legend to explicitly state this "normalized input" strategy to prevent misinterpretation.

Supplementary Figure 13. (B) PKH26-labeled tumor cell-derived extracellular vesicles (EVs) were added to the culture medium of hepatocytes. Immunofluorescence analysis demonstrated efficient uptake of these EVs by hepatocytes. EV input was

normalized across groups (10 µg per group). (C) Western blot analysis confirmed that extracellular vesicles isolated from the conditioned medium of KIAA1199-high CRC cells were more abundant than those from KIAA1199-low CRC cells.

4. Exosome purification quality control:

For purified exosomes, quality control evidence should be provided, preferably including representative images/plots. A comparison of EV quantity and morphology between the two EV groups would be better included.

Response: We fully agree with the reviewer and appreciate the emphasis on adhering to MISEV guidelines for EV quality control. In fact, we have already performed Transmission Electron Microscopy (TEM) and Nanoparticle Tracking Analysis (NTA) tests. In the revised Supplementary Figure 13, we have added a comprehensive characterization and a direct side-by-side comparison of EVs derived from KIAA1199-high and KIAA1199-low CRC cells.

1. Validation of EV Identity (Purity and Integrity):

Marker Validation: Western blot analysis confirms the enrichment of classical EV positive markers (TSG101, CD63, CD81) and the absence of the non-EV endoplasmic reticulum marker (Calnexin) in our preparations, verifying high purity.

Morphology: Representative Transmission Electron Microscopy (TEM) images show that EVs from both groups display the classic cup-shaped morphology and intact bilayer membrane structure, with no overt contaminants.

2. Comparative Analysis (Quantity and Biogenesis):

Increased Secretion (Quantity): As shown in the updated Supplementary Fig. 13B, Nanoparticle Tracking Analysis (NTA) demonstrates that the particle concentration (vesicles/mL) in the KIAA1199-high group is significantly higher than that in the KIAA1199-low group under normalized culture conditions.

Biogenesis and Secretion Driver: Consistently, Supplementary Fig. 13C reveals markedly elevated levels of TSG101 (a key component of the ESCRT machinery essential for EV biogenesis) in the KIAA1199-high group. Together, these data provide robust evidence that KIAA1199 serves as a potent driver of EV biogenesis and secretion.

3. Size Distribution:

While both groups fall within the standard exosomal size range (30-150 nm), NTA profiling reveals a subtle shift towards a smaller average diameter in the KIAA1199-high group. This may reflect a shift in the biogenesis pathway or the specific sub-population of vesicles preferentially secreted upon KIAA1199 induction.

These data have been fully integrated into the revised figure legends and Results section to satisfy the requirement for rigorous EV characterization.

Supplementary Figure 13. High KIAA1199 expression influences hepatocytes through vesicle-mediated delivery of GRN.

A. GO and KEGG enrichment of DEGs in CT26 cells: KIAA1199 overexpression vs. control (bar graph). **B.** Exosomes were isolated from the conditioned media of CT26 cells transfected with empty vector (Vector) or KIAA1199 overexpression (KIAA1199). Representative transmission electron microscopy (TEM) images show typical cup-shaped morphology of purified exosomes (upper panels). Nanoparticle tracking analysis (NTA) was performed to determine particle size distribution and concentration of exosomes from the two groups (lower panels). Scale bar, 100 nm. **C.** Western blot analysis of the enrichment of classical EV positive markers (TSG101, CD63, CD81) and the absence of the non-EV endoplasmic reticulum marker (Calnexin), these extracellular vesicles were isolated from the conditioned medium of KIAA1199 overexpression and vector CRC cells. **D.** PKH26-labeled tumor cell-derived EVs were

added to the culture medium of hepatocytes. Immunofluorescence analysis demonstrated efficient uptake of these EVs by hepatocytes. EV input was normalized across groups (10 µg per group).

5. In line 364, there was a punctuation error“either high or low levels of KIAA1199. when the primary tumor was established but before any spontaneous metastasis occurred,”

Response: We thank the reviewer for the close reading. The sentence has been amended to: "First, orthotopic tumors were established using CRC cells engineered to express either high or low levels of KIAA1199. When the primary tumor was established but before any spontaneous metastasis occurred, we introduced a standardized liver metastasis model via spleen injection of CRC cells."

REVIEWERS' COMMENTS

Reviewer #2 (Remarks to the Author):

The authors have addressed all my concerns.

Response: We thank Reviewer #2 for their positive assessment and for confirming that our revisions have satisfactorily addressed their concerns. We appreciate the time and effort they dedicated to improving our manuscript.